# Influence of W Addition on Microstructure and Resistance to Brittle Cracking of TiB_2_ Coatings Deposited by DCMS

**DOI:** 10.3390/ma14164664

**Published:** 2021-08-18

**Authors:** Edyta Chudzik-Poliszak, Łukasz Cieniek, Tomasz Moskalewicz, Kazimierz Kowalski, Agnieszka Kopia, Jerzy Smolik

**Affiliations:** 1Faculty of Metals Engineering and Industrial Computer Science, AGH University of Science and Technology, 30-059 Krakow, Poland; lukasz.cieniek@agh.edu.pl (Ł.C.); tmoskale@agh.edu.pl (T.M.); kazimierz.kowalski@agh.edu.pl (K.K.); kopia@agh.edu.pl (A.K.); 2Łukasiewicz Research Networks—Institute for Sustainable Technology, 6/10 Pułaskiego St., 26-600 Radom, Poland; jerzy.smolik@itee.radom.pl

**Keywords:** TiB_2_-W coatings, magnetron sputtering, XRD, TEM, XPS

## Abstract

The aim of this work was to determine the influence of the tungsten addition to TiB_2_ coatings on their microstructure and brittle cracking resistance. Four coatings of different compositions (0, 7, 15, and 20 at.% of W) were deposited by magnetron sputtering from TiB_2_ and W targets. The coatings were investigated by the following methods: X-ray diffraction (XRD), transmission electron microscopy (TEM), atomic force microscopy (AFM) and X-ray photoelectron spectroscopy (XPS). All coatings had a homogeneous columnar structure with decreasing column width as the tungsten content increased. XRD and XPS analysis showed the presence of TiB_2_ and nonstoichiometric TiBx phases with an excess or deficiency of boron depending on composition. The crystalline size decreased from 27 nm to 10 nm with increasing W content. The brittle cracking resistance improved with increasing content of TiBx phase with deficiency of B and decreasing crystalline size.

## 1. Introduction

One of the main research and development directions in materials science and engineering is inventing suitable coatings for different device parts in many demanding applications. This development is particularly observed in the aviation [1], energy [2], automotive [3,4], and tool industries [5]. Tool coatings greatly improve their performance possibilities, e.g., high mechanical and thermal loads, intensive wear, or corrosive environments. The new anti-wear coatings are successfully applied to cutting tools for aluminum alloy machining, engine components in aerospace applications, machine parts for die bearing, and diffusion barriers.

Anti-wear coatings are characterized by higher wear resistance and hardness. In many applications, such a high hardness is not advisable due to brittle cracking occurring in this type of coating. Titanium diboride (TiB_2_) is an attractive ceramic material because of its high thermal stability, high melting point (3000 K), high hardness (3000 kg/mm^2^), low density (4.52 g/cm^3^), high-temperature oxidation resistance (1000 °C), and high wear resistance [6,7,8,9,10,11,12,13]. However, its practical applications are still limited due to low adhesion, residual stress, and, consequently, brittle fracture. In the literature, the authors proposed that the addition of certain elements to TiB_2_ coatings can reduce residual stresses and increase adhesion, making Ti–B–M (M = Ni, Cr, Zr, C, W) coatings excellent candidates for applications in cutting tools.

H. Wang et al. [14] deposited TiB_2_ coatings with added Ni by magnetron sputtering. In the range 0 to 12.6 at.% Ni content, only TiB_2_ phase was found, with no Ni-containing crystals present in the coatings. These coatings were as hard as the pure TiB_2_ coating and could be toughened without hardness loss by Ni. E. Contreras et al. [15] analyzed the coatings TiB_2_–C. All Ti–B–C coatings exhibit high friction coefficients. This is due to the nanocrystalline structure and high hardness. However, the wear rates of all Ti–B–C coatings are significantly lower than the uncoated substrate. X. Liu et al. [16] deposited the Fe–TiB_2_ cermet coating with carbon nanotubes (CNTs) by high velocity air fuel (HVAF) spraying. With CNT addition, the Fe–TiB_2_ cermet coating shows excellent wear resistance with the lowest volume loss and specific wear rate. J. Smolik et al. [17] produced TiB_2_ coatings doped with tungsten by magnetron sputtering. In this work, the chemical composition, mechanical properties, and internal structure of the coatings have been investigated. It has been shown that doping TiB_2_ coatings with tungsten significantly changes their microstructure and brittle fracture mechanism [17].

This study aimed to investigate the effect of tungsten addition (from 0 to 20 at.%) on the microstructure and brittle cracking resistance of the TiB_2_ coatings produced by DC magnetron sputtering.

## 2. Materials and Methods

### 2.1. Coating Deposition

Ti_1−x_W_x_B_2_ coatings with different tungsten content, x = 0; 0.22; 0.42; 0.5, were deposited on a steel substrate by DC magnetron sputtering. The coatings were formed in pure Ar discharges using two circular magnetrons with targets made of pure tungsten (purity 99.95 at.%) and commercial titanium boride (purity 99.95 at.%). The targets were set at an angle of 120° to each other. Both targets were 100 mm in diameter and 7 mm in thickness. The substrate specimens were made from W320 hot work steel, tempered to a hardness of 50–52 HRC by the Rockwell method. A Seco/Warwick 10.0 VPT-series 4035/36IQ/K (Świebodzin, Poland) was used to carry out the heat treatment. After the vacuum heat treatment, the substrate samples were subjected to mechanical polishing using a RotoPol 11 polisher from Struers (Copenhagen, Denmark), until the roughness parameter Ra = 0.03 was achieved. Just before the coating deposition, the substrate samples were washed with pure alcohol. During the deposition, the pressure of Ar was kept at 5.2 × 10^−3^ mbar. To obtain TiB_2_–W coatings with three different tungsten contents, a source power of the W target was set to 25, 50, and 75 W. Additionally, a coating without tungsten was prepared for comparison. The source power of the TiB_2_ target was kept at 1000 W. All coatings were deposited at 300 °C using resistance heaters for 1 h. In this paper, we used the following shell nomenclature Ti_1−x_W_x_B_2_ where x = 0; 0.22; 0.42; 0.5.

### 2.2. Microstructure Examinations

The X-ray diffraction (XRD) method was used to identify phases present in the coatings. For this purpose, the PANanalytical Empyrean DY 1061 diffractometer (Malvern, UK) equipped with a Cu Kα tube was used. Analysis was made in GRAIZING geometry with the angle α = 1°. The XRD patterns were refined using the software Highscore Plus (PANanalytical, Malvern, UK, V5.0) suite. Determination of crystallite sizes was carried out using Scherrer’s Equation (1) [18].
(1)$$D=\frac{K\cdot \lambda }{\beta \cdot \;\cos\Theta }$$

The microstructure of the coatings on their cross-sections was studied using transmission electron microscopy (TEM) on a JEM-2010ARP (JEOL, CROISSY-SUR-SEINE, France) Plus operated at 200 kV. Samples were prepared by focused ion beam (FIB) method (Ga + ions, FEI Dual Beam) perpendicularly to the surface of the coated samples. A thin cross-section is elevated and welded to a TEM mesh using Pt sputtering.

X-ray photoelectron spectroscopy (XPS) was used to determine the chemical states of elements across the coatings, as well as their atomic concentrations. The PHI5000 Versa Probe II device (ULVAC-PHI, Chigasaki, Japan), equipped with an SCA-type analyzer, was applied. An electron pass energy for high resolution scans was equal to 46.95 eV. The monochromatic Al Kα source (photons energy 1486.6 eV) was operated at a tension of 15 kV and 4 mA emission current. Argon was used as a protective gas atmosphere, and its partial pressure was maintained at less than 3 × 10^−9^ mbar during the measurements. The binding energy scale was calibrated assuming that the position of the adventitious carbon C 1s line is equal to 284.8 eV. An argon ion Ar^+^ gun was used to sputter the coatings in order to make deeper XPS analyses. The ion gun operated at 4 keV with an ion current equal to 3 μA on area 2 mm × 2 mm with Zalar rotation. One cycle of argon sputtering lasted 20 min.

The coatings surface topography was analyzed by Atomic Force Microscopy (AFM) using a Veeco Dimension^®^Icon™ SPM (PLAINVIEW NY, USA) with the NanoScope V instrument.

CSM tester was used for nanohardness measurements with a Berkovich diamond indenter with the indenter normal force 350 mN during t = 12 min. Base on Laugier model formula (Equation (2)) [19], the value of *K_IC_* was determined.
(2)$$K_{IC}=x_{v}\cdot {\left(\frac{a}{l}\right)}^{\frac{1}{2}}\cdot {\left(\frac{E}{H}\right)}^{\frac{2}{3}}\cdot \frac{P}{c^{\frac{3}{2}}}$$

## 3. Results and Discussion

### 3.1. X-ray Diffraction (XRD)

Phase analysis was made by XRD in grazing geometry (Figure 1). The results change significantly with increasing tungsten. Peaks are widened and the intensity decreases. Analysis of diffraction patterns reveals that two phases can be identified in the coatings. Seven peaks, (001), (100), (101), (002), (102), (111), and (112), are attributed to the hexagonal TiB_2_ (No. of pattern 04-002-5184). The second phase is TB_1.94_ (No. of pattern 04-001-9272). The occurrence of this phase is visible at 2θ = 28.16° as a broadening of the (001) peak. In coatings deposited by Physical Vapour Deposition (PVD) method, the phases with highly overstoichiometric in boron (TiB_2+x_) or substoichiometric diborides (TiB_2-x_) were observed in [20].

The deficiency of boron may be due to segregation with the nanocolumnar grain boundaries during deposition and form amorphous phase. A texture was observed in deposited coatings, as shown in Figure 1a–c. The peak with maximum intensity in the pattern is at 2θ = 44.46°. For coatings, except this with the highest content of tungsten, the preferential orientation is (001) at 2θ = 27.6°. For the coating Ti_0.5_W_0.5_B_2_, the peaks are broader and the main peak is for the orientation (101) at 2θ = 44.5°. At this angle, one should also observe the main peak of the steel substrate Fe. The calculated depth of the X-ray penetration at the grazing angle α = 1° is equal to 0.35 mm. Because the thicknesses of all coatings are approximately 1 µm, this peak is identified as belonging only to the TiB_2_ phase. With increasing W content, all peaks are shifted on the right at 0.2° for Ti_0.78_W_0.22_B_2_ and at 0.27° for Ti_0.58_W_0.42_B_2_ concerning TiB_2_, as shown in Figure 1. This indicates the replacing of Ti by W. The same effect was observed in TiB_2_ doped by Si and Ta [21,22] and in coating TiWB [23]. J. Ch. Ding [21] showed that, with increases in the Si content (0–9.5% at), the peaks became broader, and their intensity decreases, which suggests that the incorporation of Si caused a decrease in the crystal size and inhibited the growth of the TiB_2_ crystal phase. The solid solution of the transition metals was also observed by A. Newirkowez et al. [24] in the coating (Ti,W,Cr)B and Z. Chluba et al. in TiTaB_2_ [22].

Based on the Scherrer equation for the peak (001), the crystalline sizes were calculated. The vales were D = 27 ± 3 nm for TiB_2_, D = 20 ± 2 nm for Ti_0.78_W_0.22_B_2_, and D = 10 ± 1 nm for Ti_0.58_W_0.42_B_2_. The results show that, with increased doping, the crystalline size decreases. Due to the high broadening and difficulties in determining the peaks for the Ti_0.8_W_0.2_B_2_ coating, the crystalline size was not determined.

### 3.2. Transmission Electron Microscopy (TEM)

To obtain structural information and the thickness of the coatings, TEM analysis on the samples’ cross-sections was performed. The TEM images in Figure 2 show cross-sectional views of Ti_1−x_W_x_B_2_ coatings. The deposited coatings have a thickness of 1.0, 1.1, 1.2, and 1.3 μm, respectively. In all samples, an amorphous sublayer between the coating and the substrate is present. The thickness of the sublayers ranged from 100 to 200 nm. All coatings have a homogeneous columnar structure, which is a typical structure for coatings deposited by PVD methods. From Thornton’s model, taking into account the deposited parameters such argon pressure and temperature T/Tm, the coating structure is typical for Zone 1 (fibrous structure). The width of the columns is in the range of 25–75 nm and decreases with the W content. The solutions of the electron diffraction patterns for each coating showed the presence of the TiB_2_ phase only (Figure 3). The reflections belonging to the (002) TiB_2_ ring are visible for all coatings. It is very strong only for TiB_2_ coating. With increases in tungsten, the reflections are blurred. This suggests decreases in the grain size. A similar result was found for TiB_2_ doped with Ni [10]. We did not observe nonstoichiometric phase TiB_1.94_ because the electron diffraction was made on a spot covering the entire coating. Nonstoichiometric phase TiB_1.94_ can be formed on grain boundary of the columns. The second reason is that the lattice parameters for TiB_2_ and TiB_1.94_ are too close (a_TiB1.94_ = 0.3053 nm, b_TiB1.94_ = 0.3053 nm, c_TiB1.94_ = 0.3157 nm, a_TiB2_ = 0.303 nm, b_TiB2_ = 0.303 nm, c_TiB2_ = 0.322 nm) and error in the calculation is bigger than the difference in lattice parameters.

### 3.3. X-ray Photoelectron Spectroscopy (XPS)

To obtain the distribution of elements and their chemical states across the coatings, X-ray photoelectron spectroscopy (XPS) coupled with an argon ion sputtering was applied. The following spectral lines were analyzed: C 1s (surface adventitious carbon), O 1s (surface passivation oxide layer), B 1s, Ti 2p, W 4d (elements of coating), and Fe 2p_3/2_ (steel substrate). Usually, the line W 4f is taken into account in XPS analysis; however, in our case, the line W 4f_3/2_ overlaps with the line Ti 3p, which impedes the interpretation of results. Figure 4 presents the results of the profiling for all samples. The profiles are plotted as an atomic concentration of the elements, calculated from the intensities of the spectral lines versus the time of sputtering. A first analysis of each sample was made on the as-received surface, and each further analysis was made after 20 min of argon ion sputtering cyclically repeated until the steel substrate was achieved, as shown in Figure 4. The Ar ion sputtering rate was not calibrated, so the depth scale could not be done in length units. The sputtering is usually very sensitive to the elemental composition. The results show, for all samples, a uniform distribution of elements across the coatings except the external surface where a presence of the oxides is evident, as well as some amounts of an adventitious carbon, which is a common contamination in XPS measurements. The oxides, which form a very thin surface passivation layer, completely disappeared after the first cycle of sputtering. 

Based on these results, a very thin interlayer rich in iron placed between the steel substrate and the coatings was observed. This agrees with TEM investigation. The interlayer was observed after different times of sputtering (Figure 4b—analysis after 160 min, Figure 4d—analysis after 200 min). 

As an example, Figure 5 presents the XPS spectra of the C 1s and O1s region for TiB_2_ coating. The analysis revealed the C 1s line on the surface and its absence at the deeper layers across the coating (Figure 4). The presence of carbon on the surface is a consequence of adsorption of contaminants from the environment. The peak at 284.8 eV is the C–C type bond and the second peak at 288.5 eV is bonded for O−C=O. Oxygen line O1s has an energy of BE = 531 − 532 eV. This energy is typical for oxygen in organic bond C=O. A lower value of band energy, BE = 529 eV, indicates the presence of metallic oxides on the surface as a passivation layer. 

For samples Ti_1−x_W_x_B_2_ (x = 0; 0.22; 0.42; 0.5), spectra of B 1s, Ti 2p, and W 4f are presented in Figure 5, Figure 6, Figure 7 and Figure 8. The lines were taken in the middle of the coatings (after 80, 120, and 140 min of sputtering, respectively). All elements are present in a metallic state, which is confirmed by the line positions on the binding energy scale: B 1s at 187.5 eV, Ti 2p_3/2_ at 454.0 eV, and W 4d_5/2_ at 243.0 eV.

The intensities of these lines were used to calculate their atomic concentrations at each analysis point. Averaged values of the concentration of elements for all coatings are gathered in Table 1.

As Table 1 shows, the coatings exhibit a changing stoichiometry with raising content of tungsten. The coating TiB_2_ has a considerable excess of boron. compared to the TiB_2_ formula. This is a characteristic effect of the magnetron deposition of the TiB_2_ coatings and already reported by Euchner H. et al. [23]. The coating Ti_0.78_W_0.22_B_2_ has only a slight excess of boron; the composition is almost stoichiometric. Meanwhile, Ti_0.58_W_0.42_B_2_ and Ti_0.5_W_0.5_B_2_ coatings exhibit an increasing deficiency in boron. Thus, one can propose the following general formula for produced coatings: Ti_1−x_W_x_B_y_, y = 2.8 for TiB_2_, y = 2.1 for Ti_0.78_W_0.22_B_2_, y = 1.8 for Ti_0.58_W_0.42_B_2_, y = 1.5 for Ti_0.5_W_0.5_B_2_.

Neidhardt et al. [25] established the basic mechanisms to explain the presence of boron excess in TiB_2_ coatings deposited by magnetron sputtering. Using the experimental results combined with the simulations of dynamic ion transport in Monte Carlo simulations, they showed that, due to the differences in atomic weights between Ar ions and elements such as Ti and B (m_Ar_ = 39.9 u, m_Ti_ = 47.9 u and m_B_ = 10.8 u), the argon sputtered B atoms are ejected along the target normal, while the Ti atoms are ejected with an under-cosine distribution (lower angles) [26]. As a result of magnetron sputtering, the coatings contain more B than Ti. By increasing the pressure in the chamber, changing the plasma density, changing the angle between substrate and target, or changing the distance from the target to the substrate, we can reduce the Ti deficiency.

In our research, by increasing the power of the magnetron with the W target, the plasma density changes. A strong source of power influences high contents of W and high plasma density, which results in a decrease in the deficiency of B in the coatings Ti_0.78_W_0.22_B_2_ and Ti_0.58_W_0.42_B_2_.

The nonstoichiometric of TiB_y_ phase with deficiency of B can be formed as B-rich tissue amorphous phase separating stoichiometric TiB_2_ nanocolumnar structures. Another explanation is given by J. Thörnberg et al. [27] who, in coating TiB_2_ deposited by high-power impulse magnetron sputtering (HiPIMS), identified tightly packed TiB_2_ nanocolumnar structures with planar defects of Ti-enriched stacking faults, accommodating the B deficiency.

### 3.4. Atomic Force Microscopy (AFM) and Brittle Cracking Resistance (K_IC_)

The surface topography was analyzed by AFM over an area of 100 nm × 100 nm. The results are shown in Figure 9. The effect of tungsten doping on the surface topography is weakly visible. The roughness parameters Rmax, Ra, and Rq, are shown in Table 2. The values of the parameters are almost the same. The decreasing of columnar width does not influence roughness parameters. 

The brittle cracking resistance test was performed by measuring the length of cracks in the corners of the indentations made with a Berkovich indenter at a load of 350 mN, based on the Laugier formula. The results of the brittle cracking resistance are presented in Table 2 as a parameter K_IC_. Representative SEM images are shown in Figure 10. The presence of radial cracks in TiB_2_ coating at lower tungsten content coating indicated brittle failure. No radial cracks were observed in coatings with a higher content of tungsten. This indicates greater fracture toughness. The circular cracks visible in the coatings may be related to the greater hardness of the samples. Increasing tungsten content increases brittle cracking resistance. This effect can be explained by two reasons. First is the presence of nonstoichiometric phase TiB_1.94_ in coating, and the second is a decrease in crystalline size with increasing content of W. The highest brittle cracking resistance in thin films TiB_x_ was observed by Thörnberg et al. [27]. He showed that, by changing the stoichiometry in TB_2_ coating, we can influence brittle cracking resistance. The highest brittle cracking resistance was observed in coating TiB_1.43_ and the lowest in TiB_2.7_. In our case, TiB_2_ coatings were deposited only by DCMS at constant argon pressure. The magnetron power for the target TiB2 was constant, but the magnetron power for target W was changed during the process. Changing the power of the tungsten magnetron affects the plasma density and chamber pressure, which could have resulted in a similar effect as in work [27], i.e., nonstoichiometric Ti–W–B. Smolik et al. [19] explained this effect by nanostructures of coatings, which is a consequence of a change in the direction of cracking. By changing the direction of cracking, the energy of a single crack is reduced and can disappear.

## 4. Conclusions

The studies presented in this work have shown that the addition of tungsten had a considerable impact on the microstructure and the stoichiometry of the W-doped TiB_2_ coatings deposited by the magnetron sputtering on the steel substrate. Four coatings of different W content were examined: 0, 7, 15, and 20 at.%. The coating thicknesses were 1.0, 1.1, 1.2, and 1.3 μm, respectively. TEM cross-section images revealed a homogenous columnar structure of the coatings and the presence of an amorphous sublayer between the coatings and the substrate. The sublayers’ thicknesses were from 100 to 200 nm, and were rich in Fe coming from the substrate (30–50 at.%). XPS analyses combined with an argon ion profiling showed the uniform distribution of Ti, W, and B across the coatings. The important conclusion from the XPS quantitative analysis is that the TiB_2_ phase is not stoichiometric with an excess or deficiency of boron. With the addition of tungsten, the relative quantity of boron strongly decreased. XRD results showed the occurrence of the TiB_2_ and TiB_1.94_ phases for coating with tungsten content. The grain size decreased from 27 nm for the tungsten free coating to 10 nm for the coating with the 15 at.% W concentration. In this coating, two effects have a strong impact on the improvement of highest brittle cracking resistance: nanostructure of coating and presence of nonstoichiometric phase with deficiency of boron. 

## Figures and Tables

**Figure 1 materials-14-04664-f001:**
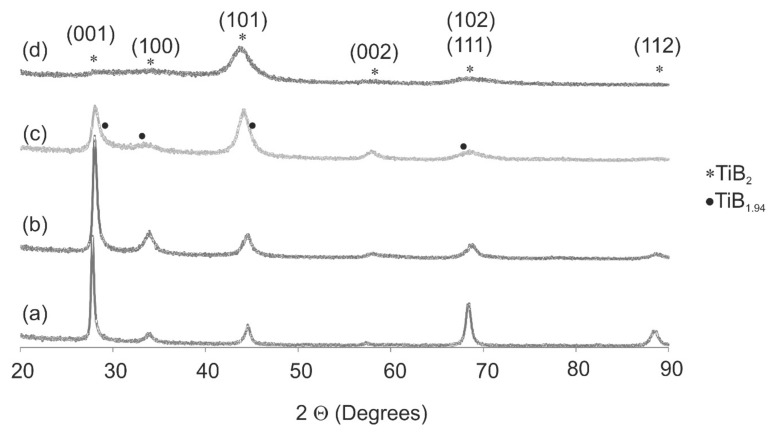
X-ray diffraction patterns for coatings Ti_1−x_W_x_B_2_ (**a**) x = 0, (**b**) x = 0.22, (**c**) x = 0.42, and (**d**) x = 0.5.

**Figure 2 materials-14-04664-f002:**
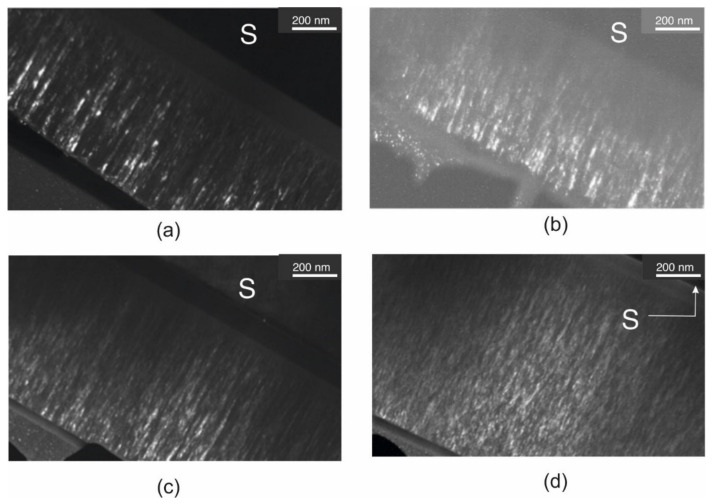
TEM images of coatings cross-sections for samples: (**a**) TiB_2_; (**b**) Ti_0.78_W_0.22_B_2_; (**c**) Ti_0.58_W_0.42_B_2_; and (**d**) Ti_0.5_W_0.5_B_2_.

**Figure 3 materials-14-04664-f003:**
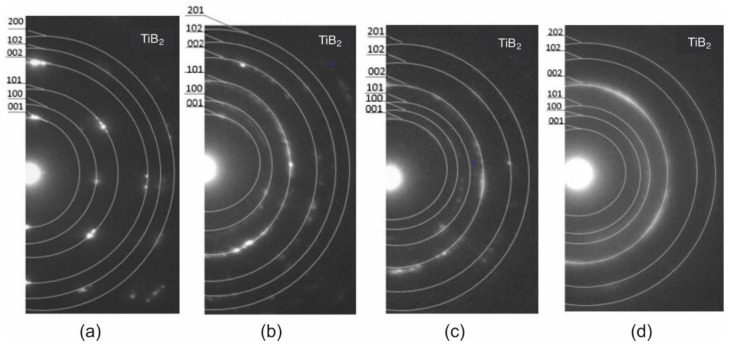
SAED patterns indicating TiB_2_ as the only phase (**a**) TiB_2_; (**b**) Ti_0.78_W_0.22_B_2_; (**c**) Ti_0.58_W_0.42_B_2_; and (**d**) Ti_0.5_W_0.5_B_2_.

**Figure 4 materials-14-04664-f004:**
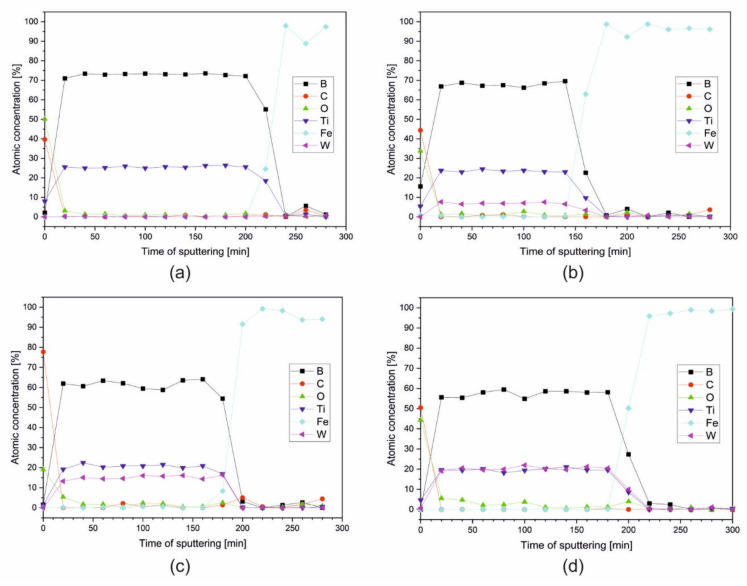
Atomic concentration depth profiles for coatings (**a**) TiB_2_; (**b**) Ti_0.78_W_0.22_B_2_; (**c**) Ti_0.58_W_0.42_B_2_; and (**d**) Ti_0.5_W_0.5_B_2_ obtained using XPS coupled with Ar ion sputtering.

**Figure 5 materials-14-04664-f005:**
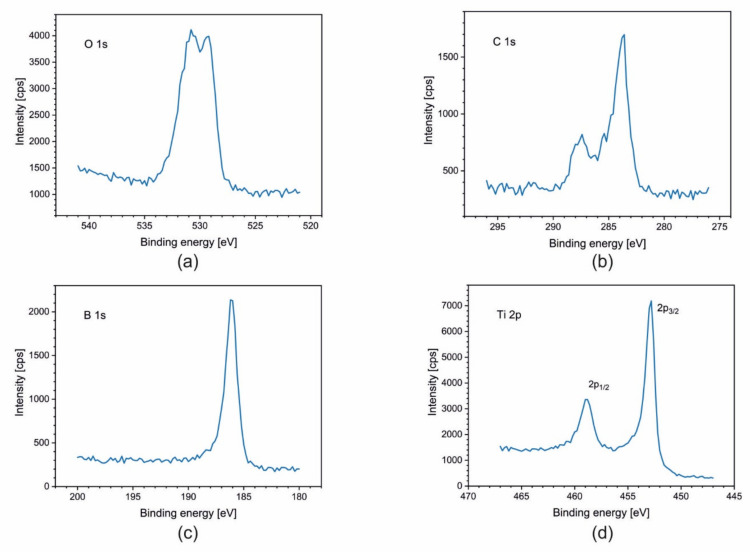
XPS spectra of (**a**) O1s; (**b**) C1s from the surface; (**c**) B 1s; and (**d**) Ti 2p obtained after 140 min of sputtering for the coating TiB_2_.

**Figure 6 materials-14-04664-f006:**
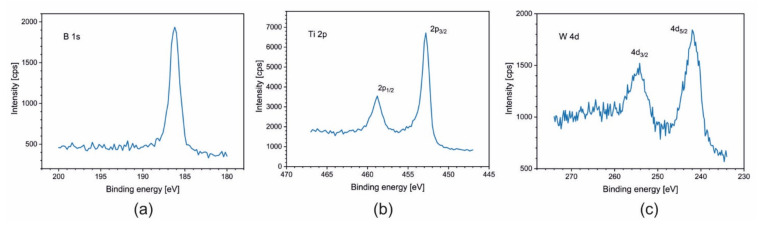
XPS spectra of (**a**) B 1s; (**b**) Ti 2p; and (**c**) W 4d obtained after 80 min of sputtering for the coating Ti_0.78_W_0.22_B_2_.

**Figure 7 materials-14-04664-f007:**
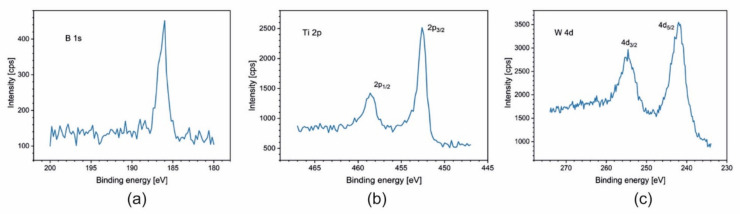
XPS spectra of (**a**) B 1s; (**b**) Ti 2p; and (**c**) W 4d obtained after 120 min of sputtering for the coating Ti_0.58_W_0.42_B_2_.

**Figure 8 materials-14-04664-f008:**
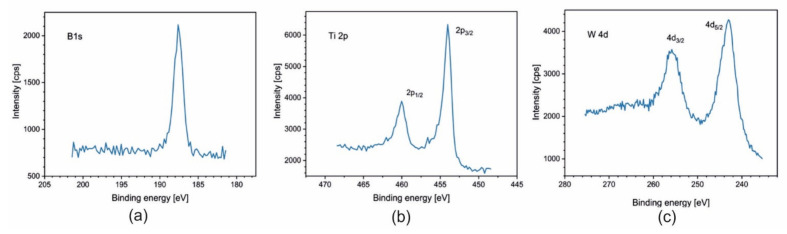
XPS spectra of (**a**) B 1s; (**b**) Ti 2p; and (**c**) W 4d obtained after 140 min of sputtering for the coating Ti_0.5_W_0.5_B_2_.

**Figure 9 materials-14-04664-f009:**
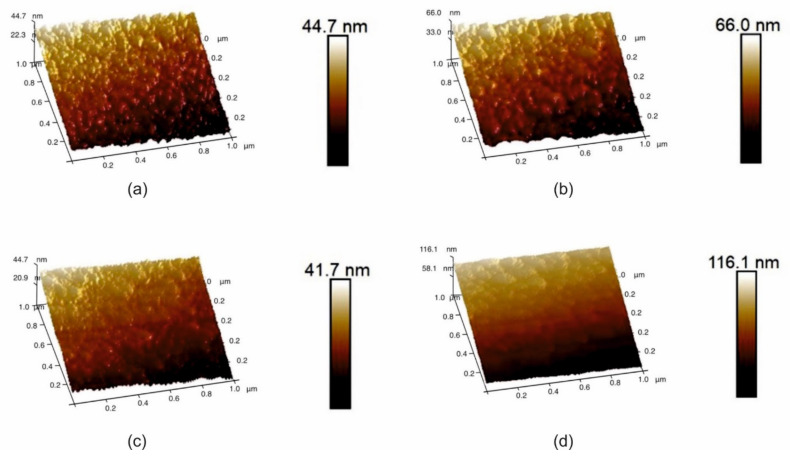
Surface morphology for samples (**a**) TiB_2_; (**b**) Ti_0.78_W_0.22_B_2_; (**c**) Ti_0.58_W_0.42_B_2_; and (**d**) Ti_0.5_W_0.5_B_2_.

**Figure 10 materials-14-04664-f010:**
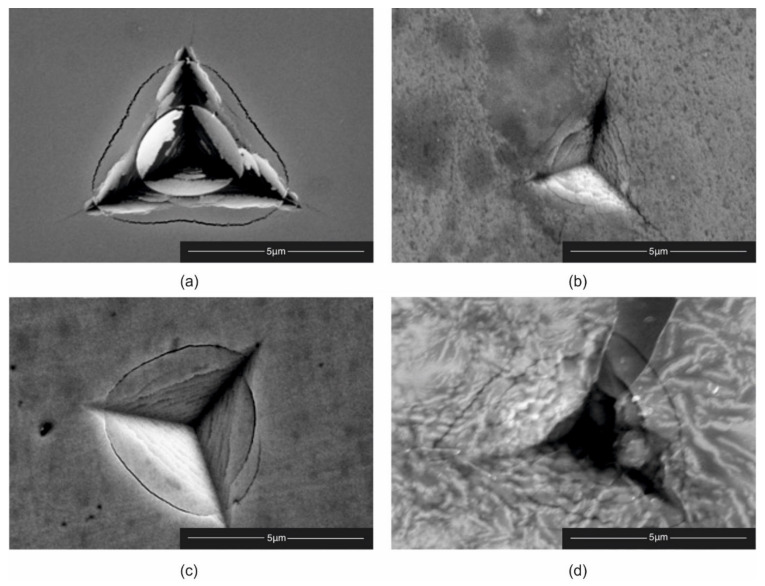
SEM images of the Berkovich indenter with load of 350 mN for *K*_IC_ for: (**a**) TiB_2_; (**b**) Ti_0.78_W_0.22_B_2_; (**c**) Ti_0.58_W_0.42_B_2_; and (**d**) Ti_0.5_W_0.5_B_2_.

**Table 1 materials-14-04664-t001:** The concentrations of elements across the coatings determined by XPS.

Sample	Ti [at.%]	W [at.%]	B [at.%]	B/[Ti + W]
TiB_2_	26	0	74	2.8
Ti_0.78_W_0.22_B_2_	25	7	68	2.1
Ti_0.58_W_0.42_B_2_	21	15	64	1.8
Ti_0.5_W_0.5_B_2_	20	20	60	1.5

**Table 2 materials-14-04664-t002:** Roughness and fracture parameters of coatings determined by AFM.

Sample	Ra [nm]	Rmax [nm]	Rq [nm]	K_IC_
TiB_2_	1.58 ± 0.15	11.1 ± 1.1	1.99 ± 1.1	1.8
Ti_0.78_W_0.22_B_2_	1.46 ± 0.15	12.9 ± 1.2	1.81 ± 1.1	2.8
Ti_0.58_W_0.42_B_2_	1.03 ± 0.1	8.10 ± 0.8	1.25 ± 1.1	8.5
Ti_0.5_W_0.5_B_2_	1.34 ± 0.13	8.9 ± 0.9	1.63 ± 1.1	11.5

## Data Availability

Not applicable.

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
