# Peer review of "Influence of W Addition on Microstructure and Resistance to Brittle Cracking of TiB2 Coatings Deposited by DCMS"

_materials, 2021, doi:10.3390/ma14164664_

Round 1

Reviewer 1 Report

In the manuscript presented, authors prepare and study Ti-W-B coatings in steel substrates. Investigation of super hard coatings based on light elements is actual topic for broad technoligies. Systematic studies will give fundamental information to rationally design new materials and find them applications. Study is clearly organised and contains all necessary experimental information. Nevertheless, before connsidering for publication, several issues should be clarified.

In general, films compositions were mentioned already at the beginnig of the manuscript, nevertheless, XPS study appeares only later on. I suggest to discuss films composition at the early beginnig. It seems that composition of the coating is quite uniformal, which shuld be additionally stressed in the text. Nevertheless, according to Table 1 the composition of the last sample should be Ti0.67W0.67B2, but it "nominal" composition is reported as Ti0.8W0.2B2. I suggest to revise lines 198-204 to give more clear explanation for actual composition as well as nominal compositions used in the text.

Line 95: diffraction patterns (not "spectra").

Line 97: Authors suggest a presence of minor TiB1.94 phase in the coating. Nevertheless, there is no individual diffractoin line corresponding to the phase mentioned were detected. Anisotropic broadening of the characteristic for TiB2 phase 001 line can be described without a presence of second phase due to presencee of planar defects etc. Presence of second phase is not convincing from the current XRD data. For me, both PDF cards correspond to the same phase with slightly different cell parameters and (probably) different composition. If "TiB2" and "TiB1.94" correspond to two significantly different phases their crystal structures as well as characteristic XRD patterns should be described and discussed in more details.

Lines 113-114: I suggest to give also cell parameters and corresponding experimental errors for cristallite sizes and cell parameters.

Lines 223-224 and capture for Table 2: I suggest to give explanation for parameters refined.

Line 232: "resista" should be replaced by "resistance".

References in the text such as line 240 should be revised: "J. Thörnberg and all." should be replaced with "Thörnberg et al."

Reviewer 2 Report

There are some issues with this paper.

One issue is the film composition. The films are described in the text as Ti0.93W0.07B2, Ti0.85W0.15B2,Ti0.8W0.2B2 and TiB2. However, according to Table 1, the actual Ti:W ratios are Ti0.78­W0.22B2.1, Ti0.58W0.42B1.8, T­0.5W0.5B1.5  and TiB2.8. I accept that the B concentration is nominally B2 (more on this later) but where do the Ti:W ratios in the text come from?

Regarding the nominally Bcontent, the changes in fracture toughness are ascribed to the changing W content, however, the authors themselves have referenced a paper [27] where change in stoichiometry alone gave rise to changing fracture toughness. Can the authors disentangle the two effects?

How did the film thickness vary with W content? Was it systematic? Film thickness measurements need to be given. In fig 4 showing the elemental profiles, the time taken to sputter through the films varies from 160 to 200 min, Is this due to film thickness variations or differences in Ar etching rate?

In fig 5b showing the C1s peak, The paper states that XPS energies were calibrated by the adventitious C peak at 284.8 eV. In that case, what are the origin of Th peaks at approx. 283.5 and 287.5 eV? A deconvolution of the curve would be helpful here.

In the discussion of the XRD results, the authors state in lines 102-103 “the preferential orientation is

(001) at 2θ=27.6°. They should make clear that the (001) planes are not parallel to the substrate surface since this was measured with grazing incidence. Also, the curve in fig 1 for TiB2 shows additional peaks at approx. 65 and 82 degrees 2theta. What are these due to?

Target configuration: given the 120o angle between the targets, how did the authors ensure that the composition was uniform across the substrates?

What was the bias conditions of the substrate? Earthed, biased or floating?

There are some typographical and grammatic errors. I have listed some here:

Line 71 “Scherer's” should be “Scherrer's”

Line 86 “intender” should be “indenter”

Line 90 The parameters should be specified in Equation (1)

Line 97 “occur” should be “occurrence”

Line 127 “what is typical structure” should be “which is a typical structure”

Lines 133-134 “TiB2” should be “TiB­1.94

Line 135 “to” should be “too”

Line 151 “what impedes” should be “which impedes”

Line 165 “Base on this results” should be “Based on these results”

Line 232 “resista” should be “resistance”

Reviewer 3 Report

You have indicated some industry which can be used as target applications; however it will be much better if the authors could indicate some typical components coated with these solutions.

“wear resistance and hardness”..I assume here higher hardness ?

Some small English errors were noted everywhere please try to avoid them , an example here “But its practical applications are still limited”

Please avoid block citation as [6-14]; other wise it is noted reference 7 and 13 are the same ! My suggestion is to provide critical evaluation for each reference; otherwise apart of citing some references from literature and saying what they made is nothing critical and the reviewer do not understand what is the weakness from these work and why it is necessary this work  

The authors contribution to this research field is not highlighted

Also there is required to explicitly indicate the scientific novelty of this work

Why these condition was selected  “x= 0; 0.07; 0.15; 0,20” and not lower amount or higher ?

The polishing procedure is very evasive not difficult to replicate please provide details

Please provide a ref for “using Scherer's equation”

“Samples were prepared using a Focused Ion Beam (FIB) method.” Again, this is very evasive detail!

“The results change significantly with the increase of tungsten content, as shown in Fig. 1.” Please provide an interpretation in text of the results provided in Figure 1 cause indeed there are some changes but not obvious one; also it will be much better making also description against literature data, how these evolves !

Provide clear indication where this was observed, is this in a Figure, if so in which Figure “A texture was observed in deposited coatings”

“the crystalline size wasn’t determined” OK but if this was a target in this research how do you justify scientifically ? cause this information could be very important

“In all samples an amorphous sublayer between the coating and the substrate is 125

present.” Probably you are right but there is no evidence !

In which Figure was presented this information “A first analysis of each sample was made on the as-received surface and each further analysis was made after 20 min of argon ion sputtering cyclically repeated till the steel substrate was achieved.”?

“are presented in Fig. 5-8.” It will be much better discussion these Figures individually

Round 2

Reviewer 1 Report

Authors addressed all comments.

Reviewer 2 Report

Minor grammatical error :-

Line 109  “shift on the right at” should be “shifted to the right by"

Reviewer 3 Report

-